# The Effect of Humic Substances as an Organic Supplement on the Fattening Performance, Quality of Meat, and Selected Biochemical Parameters of Rabbits

**DOI:** 10.3390/life12071016

**Published:** 2022-07-08

**Authors:** Zuzana Lacková, František Zigo, Zuzana Farkašová, Silvia Ondrašovičová

**Affiliations:** 1Department of Nutrition and Animal Husbandry, University of Veterinary Medicine and Pharmacy, 041 81 Košice, Slovakia; zuzana.lackova@uvlf.sk (Z.L.); zuzana.farkasova@uvlf.sk (Z.F.); 2Department of Biology and Physiology, University of Veterinary Medicine and Pharmacy, 041 81 Košice, Slovakia; silvia.ondrasovicova@uvlf.sk

**Keywords:** rabbits, humic substances, carcass value, biochemical parameters, cholesterol

## Abstract

In this study, we assessed the effect of humic substances, as an organic supplement in feed, on the fattening performance, meat quality and selected lipid and mineral parameters from the blood serum of rabbits. Three groups of the Giant Saris rabbit breed were used (one control and two experimental), with 16 animals per group. The animals in the control group were fed a standard pellet diet, the humic substances group received a basal diet supplemented with 5% humic substances, and the third group received a basal diet with 5% humic–fatty substances preparation during the entire experiment (from 35 to 120 days of age). There were 85 days of fattening; then, the rabbits were slaughtered. In the group supplemented with 5% humic–fatty substances addition, we noticed a higher final weight (*p* < 0.05) and higher average daily gains compared to the control group at the end of the fattening period, at 120 days of age. On the other hand, a slightly lower final weight (*p* > 0.05) in the group supplemented with humic substances was found compared to the control group. In the comparison of the individual parameters of the meat quality in rabbits, we observed a positive effect in the reduction in the intramuscular fat content and the lipid parameters as well as a lower total cholesterol from the blood serum in both supplemented groups. Regarding the mineral parameters, we observed elevated blood serum values of calcium and phosphorus in both experimental groups. The addition of humic–fatty substances appears to be the most effective way of supplementing rabbit feed due to the synergistic effect of humates and vegetable oils for their optimal growth development and the production of reduced-fat meat.

## 1. Introduction

Livestock farming for meat production is one of the main sectors of livestock production. In addition to traditional species such as pig, cattle, and poultry, this industry is developing by farming new species such as rabbits [1]. Rabbits are bred mainly for meat and fur, but they are also used as laboratory animals. The preference for rabbit farming is justified by the high nutritional quality and low-calorie meat, and the fact that it is unpretentious and sufficiently adaptable to extensive farms with a subsistence function [2].

In order to improve the immunity and production characteristics of livestock, in recent years, feed mixtures have started to be enriched with various additives, helped by several legislative changes in Europe, linked to the elimination of the use of antibiotics, growth promoters, and other chemical preparations in livestock nutrition [3]. One possible alternative is the addition of humic substances (HS), which are among the most widespread natural organic compounds. Humates are formed by the chemical and biological decomposition of organic substances, especially plants and animals. Due to their unique composition and excellent properties, they are able to act complexly in animals, thereby increasing an animal’s defenses against disease and stimulating higher performance [3,4].

The aim of this work was to compare the impact of the addition of two products based on humic substances (HS) and humic–fatty substances (HFS) in a granular compound feed on the fattening performance, quality of meat, and selected biochemical parameters in rabbit blood serum.

## 2. Materials and Methods

### 2.1. Placement of Rabbits

The rabbits were bred under the same conditions in an air-conditioned hall in standard cages for rabbit breeding (80 × 40 × 66 cm); in each cage, there were four rabbits. During the experiment, light was provided for 16 h, with 8 h of darkness. The average temperature in the hall was 22 ± 4 °C, and the relative humidity was 70 ± 5%. The experiment was performed after approval from the Ethics Commission of the University of Veterinary Medicine and Pharmacy in Košice (protocol code EKV/2022-11 and the date of approval was 16 May 2022).

### 2.2. The Selection of Rabbits and Feed Supplementation

The study included 48 rabbits of the Saris giant rabbit breed of both sexes at the age of 35 days, divided into three groups, namely a control group without the addition of HS and two groups with 5% addition of HS or HFS. The selection of rabbits into groups was random; sixteen rabbits were selected for each group, and each group consisted of 4 replicates, with four rabbits within each replicate.

The control group was given a pellet basal diet for growing rabbits without any supplements, according to the recommendations given in the international feeding standards [5], containing dried alfalfa 17%, dried grass 11%, wheat bran 13%, wheat oat 5%, wheat grain 10%, barley grain 10%, corn 15%, extracted soybean meal 8%, sugar beet pulp 2%, rape meal 5%, rapeseed oil 2%, and a mineral–vitamin premix 3% (containing per kg: vitamin A, 10,000 IU; vitamin D3, 1000.0 IU; vitamin E, 700 mg; vitamin K, 30.0 mg; Mn, 40 mg; Cu, 10 mg; Fe, 50.0 mg; Zn, 65 mg; Co 0.2 mg; iodine, 0.638 mg; Se, 0.2 mg; and F, 0.032 mg). The diet contained 10.6 MJ/kg metabolizable energy, 6.8% crude ash, 15.4% crude protein, 16.7% crude fiber, 4.7% crude fat, 19.5%, starch, 7.2% soluble sugars, 0.70% lysine, 0.31% methionine, 0.90% calcium, 0.55% phosphorus, and 0.25% sodium.

The first experimental group was administered a pellet basal diet with 5% addition of humic substances (Humac^®^ Natur AFM; 65% HS share in the dry matter) during the entire fattening period (from 35 to 120 days of age). The second experimental group was given a basal diet with 5% addition of a humic–fatty substances preparation (HFS; Humobentofet, Tronina PHW, Wroclaw, Poland) composed of 80% humic-mineral carrier, and 20% vegetable oils (48% oleic, 20% linoleic, 5% linolenic, and 15% palmitic acid).

### 2.3. Growth Performance and Carcass Yield of Rabbits

Throughout the research period (85 days), the animals were fed the granular mixtures ad libitum and had free access to drinking water. The weight of the rabbits was determined by weighing at the ages of 35, 60, 90, and 120 days. At the end of the experiment, after 12 weeks of HS and HFS administration, at 120 days of age, 6 male rabbits from each group were randomly selected for blood and meat collection. The selected rabbits were killed by electric stunning (STZ-6, Koma, Bratislava, Slovakia) with a minimum current of 0.3 A and a voltage of 110 V, with subsequent cutting through the *vena jugularis* and bleeding. The carcass weight was recorded after slaughtering (decapitation, removal of skin and distal parts of limbs, and evisceration). The yield of the carcass was determined as the ratio of the final body weight at 120 days of fattening to the carcass weight.

### 2.4. Determination of Meat and Biochemical and Slaughter Parameters

Blood samples were collected in tubes and centrifuged for 15 min to obtain blood serum at 1180 g. The chemical analysis of the basic components of meat was evaluated from dorsal muscle samples (*musculus longissimus dorsi*). The meat sample was collected 1 h after slaughter, wrapped in foil, and stored at 4 °C for 24 h until quality analysis. The pH of the meat samples was analyzed with a digital InoLab pH meter (Wissenschaftlich-Technische Werkstatten, Weilheim, Germany). The water content was determined by oven-drying at 105 °C [6]. The crude protein content was determined on a Kjeltec Auto type 1030 analyzer (Hanon, Jinan, China). The lipids were isolated in ground samples with petroleum ether in a Soxhlet apparatus (LTHS 500, Brnenska Druteva v.d., Brno, Czech Republic) and were detected according to the method in Semjon et al. [7]. The electrical conductivity was determined using an LF-STAR apparatus (Germany). The water holding capacity (WHC) was evaluated using the Grau–Hamm method based on the percentage of water remaining in 300 mg of the sample subjected to 2-kg pressure, following Joo [8]. The energy value (EV) was calculated according to the following equation:EV (Kj·100g^−1^) = 16.75 × protein content + 37.65 × fat content.

For the evaluation of the biochemical parameters, 10 mL of blood was collected from the selected rabbits before slaughter. Total cholesterol levels (TC), triglycerides (TG), and concentrations of HDL (high density lipoprotein) and LDL (low density lipoprotein) were determined from the blood serum, according to the methods in Hammand et al. [9]. Calcium (Ca), phosphorus (P), and iron (Fe) values were determined from the blood serum according to the methods in Jaďuttová et al. [4].

### 2.5. Statistical Analysis

The measured values were evaluated by one-factor analysis of variance (ANOVA) with the level of significance set at *p* < 0.05. The significance of the differences between the different growth rates, meat quality, and biochemical parameters was confirmed by means of Tukey’s multiple comparison test. This test was applied to find the means that were significantly different from each other between the monitored groups. The results in the tables are given as the average value (X) and the mean standard deviation (SD).

## 3. Results and Discussion

Table 1 presents the basic parameters of the performance of the rabbits during the fattening. When assessing the intensity of the live weight growth, we recorded a higher final weight of 3.37 kg and average daily gains of 32.8 g·d^−1^ in the HFS supplemented group, compared to the control (31.1 g·d^−1^) and HS groups (3.08 g·d^−1^) at the end of the fattening period, at the age of 120 days. Moreover, the higher weight in the HFS group at 85 days of fattening had a positive effect on the increased carcass weight and yield after slaughtering. On the other hand, in the group of rabbits with pure 5% HS addition, there were slightly lower daily weight gains, which was reflected in the lower weight (*p* > 0.05) at the end of the fattening period. 

As in our study, Mista et al. [10] reported a positive effect on higher weight gains and a higher feed conversion ratio in supplemented rabbits with a humic fatty acid preparation after 6 weeks of feeding compared to the control group, which was fed a basal diet without HS administration. Ondruška et al. [3] achieved similar results in their study, where the rabbits supplemented with humic substances with a probiotic had a higher live weight in the last phase of the fattening period (77 days) compared to the control group supplemented only with the probiotic preparation. In addition, both authors did not record statistical differences in the use of the enriched feed mixtures in the carcass yield and the meat quality indicators (moisture, total protein, and intramuscular fat). In contrast, our study confirmed a lower intramuscular fat in the meat of both groups of rabbits supplemented with 5% HS or HFS (Table 2).

**Table 1 life-12-01016-t001:** Comparison of the weight, average daily gain, and carcass value during the rabbit fattening period.

Group/Fattening	*n*	Control	HS	HFS
		X ± SD	X ± SD	X ± SD
35 days (kg)	6	0.61 ± 0.21	0.60 ± 0.13	0.58 ± 0.17
60 days (kg)	6	1.10 ± 0.28	1.06 ± 0.25	1.15 ± 0.32
90 days (kg)	6	2.26 ± 0.22	2.14 ± 0.28	2.3 ± 0.35
120 days (kg)	6	3.15 ± 0.14 ^a^	3.08 ± 0.19 ^a^	3.37 ± 0.21 ^b^
Average daily gain g·d^−1^	6	31.1	29.1	32.8
Carcass weight (kg)	6	1.81 ± 0.12 ^a^	1.75 ± 0.09 ^a^	1.98 ± 0.15 ^b^
Carcass yield (%)	6	57.1	56.7	58.5

Note: *n*—number of samples from each group; HS—group of rabbits supplemented with 5% humic substances preparation in basal die; HFS—group of rabbits supplemented with 5% humic–fatty substances preparation in basal diet; ^a,b^—values in rows with different letters differ significantly at *p* < 0.05.

The slightly decreased body weight and intramuscular fat in the meat in the HS group may be due to the increased fat metabolism from the beneficial detoxifying effects of the humic substances on the liver. Due to the increased fat metabolism, lower blood cholesterol levels were observed in both supplemented groups compared to the control group. In addition to lower cholesterol, decreased levels of LDL-cholesterol and triglycerides in blood serum were confirmed in the HS group (Figure 1). Mista et al. [10] also found a cholesterol reduction in rabbit blood serum, in particular, in the LDL fraction.

**Table 2 life-12-01016-t002:** Comparison of the individual parameters of meat quality in rabbits.

Parameter	*n*	Control	HS	HFS
		X ± SD	X ± SD	X ± SD
pH	6	5.55 ± 0.1	5.45 ± 0.05	5.50 ± 0.05
Moisture%	6	75.60 ± 0.65	74.60 ± 0.91	75.40 ± 0.75
Total proteins%	6	22.40 ± 0.35	22.8 ± 0.25	23.3 ± 0.51
Intramuscular fat%	6	2.08 ± 0.13 ^a^	1.4 ± 0.26 ^b^	1.8 ± 0.20 ^c^
WHC%	6	34.6 ± 2.63	33.8 ± 2.01	34.1 ± 2.18
Electrical conductivity (μS)	6	0.70 ± 0.15	0.80 ± 0.10	0.70 ± 0.20
Energetic value (kJ)	6	436.3 ± 14.5	409.4 ± 14.5	448.2 ± 17.50

Note: *n*—number of samples from each group, HS—group of rabbits supplemented with 5% humic substances preparation in basal diet; HFS—group of rabbits supplemented with 5% humic–fatty substances preparation in basal diet. WHC—water holding capacity; ^a,b,c^—values in rows with different letters differ significantly at *p* < 0.05.

Calcium and phosphorus are essential macroelements for bones, as well as for energy metabolism. Supplementation of HS and HFS in our experiment was associated with an increase (*p* < 0.05) in calcium and phosphorus in the blood serum of both supplemented groups of rabbits (Figure 1). Bahadori et al. [11] and Ozturk et al. [12] also observed an increase in Ca and P in the blood of broiler chickens after supplementation with humic substances. Conversely, Rath et al. [13] and Jaďuttová et al. [4] detected a decrease in both Ca and P in blood after the addition of HS to feed for broilers, which may be due to the ability of HS to chelate metals affected by a large number of carboxylic acid side chains [13]. On the other hand, Arif et al. [14] did not notice any changes in the levels of these elements in the blood of quails. The effect of HS on the concentration of Ca and P in the blood may be influenced by the amount added to the feed.

## 4. Conclusions

Based on the synergistic effect of humic substances and vegetable oils, there was a higher final weight and carcass yield in the group of rabbits supplemented with a 5% addition of HFS in the feed during 85 days of fattening. In addition, supplementation with HS or HFS positively affected the lipid and mineral parameters, which was manifested by lower total cholesterol as well as increased levels (*p* < 0.05) of Ca and P in the blood serum of both supplemented groups. These results make it possible to assume that the administration of humic–fatty substances in rabbit nutrition would improve meat quality indicators and biochemical parameters in the blood due to their synergistic effect. A positive effect can be assumed mainly with the reduction in the fat and cholesterol levels and with the increase in the serum concentrations of essential trace elements, especially Ca and P.

## Figures and Tables

**Figure 1 life-12-01016-f001:**
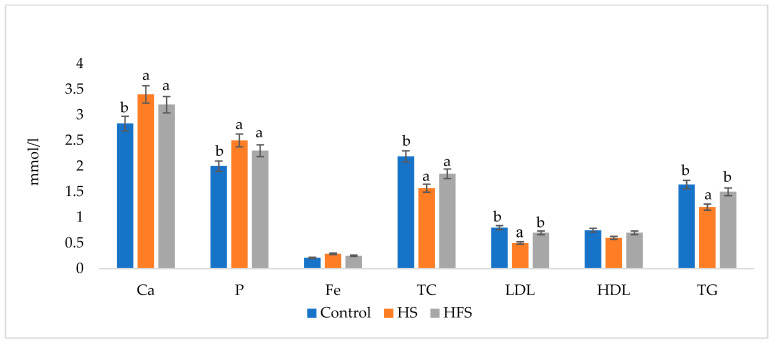
Comparison of selected biochemical parameters in the blood serum of the rabbits. Note: Ca—calcium. P—phosphorus. Fe—iron. TC—total cholesterol. LDL—low density lipoprotein. HDL – high density lipoprotein. TG—triglycerides. HS—group of rabbits supplemented with 5% humic substances preparation in basal diet. HFS—group of rabbits supplemented with 5% humic–fatty substances preparation in basal diet. ^a,b^—values over the columns with different letters differ significantly at *p* < 0.05.

## Data Availability

Not applicable.

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
