# Peer review of "The Effect of Humic Substances as an Organic Supplement on the Fattening Performance, Quality of Meat, and Selected Biochemical Parameters of Rabbits"

_life, 2022, doi:10.3390/life12071016_

Round 1
Reviewer 1 Report
Dear Editor and Authors,
I send you my review about the paper “The effect of humic substances as an organic supplementation on the fattening performance, quality of meat and selected biochemical parameters of rabbits”.
The paper was aimed to to compare the impact of the addition of two products based on humic substances (HS), and humic-fatty substances (HFS) in a granular compound feed on the fattening performance, quality of meat, and selected biochemical parameters in rabbit blood serum.
As a general comment, the paper is interesting; containing a novelty and the text is well written.
The results can be beneficial in finding a suitable replacement for antibiotics as an addition of feed to improve health status, carcass parameters and quality of rabbit meat.
The results, is well presented and adequately discussed to support the aim of the paper, however, in the tables should be reported the number of samples.
I noticed some grammatical errors in the text (Line #. 21, 26, 93), which I marked directly in the text in the attached article. I am also not clear about the content of individual acids in the HFS-based preparation (Line # no. 73), needs more explanations.
Conclusion of the paper is well written, supported by the data and summarized the most significant results and the recommendation resulting therefrom.
→ I endorse the article for publication after making these concerns.

Author Response
Thank you for your comments and suggestions that helped improve the quality of the article. We have accepted all your comments, which we have incorporated into the text.
Reviewer: In the tables should be reported the number of samples.
Response: Number of samples was reported.
Reviewer: I noticed some grammatical errors in the text (Line #. 21, 26, 93), which I marked directly in the text in the attached article.
Response: Text was grammatically corrected (Lines #. 21, 26, 93).
Reviewer: I am also not clear about the content of individual acids in the HFS-based preparation (Line # no. 73), needs more explanations.
Response: Content of individual acids in the HFS-based preparation was corrected. (Line # no. 73).

Reviewer 2 Report
I include a file with some suggestions for authors

Author Response
Dear reviewer,
thank you for your comments and suggestions that helped improve the quality of the article. We have accepted all your comments, which we have incorporated into the text.
Abstract
Reviewer: 16-18 lines. It’s possible to avoid describe every biochemical parameter, because they must be defined in methodology.
Response: We have omitted a description of each biochemical parameter in the abstract as recommended by the reviewer. All lipid and minerals parameters from blood serum are defined in the methodology.
Reviewer: 23-26 lines. There two sentence contradictory, first one indicates final weight is higher and the other one indicates is slightly low. Table 1 indicates differences at 120 d of fattening (final weight).
Response: We corrected sentences in abstract according results presented in table 1. and we changed statistical method according to the reviewer's recommendation. We used Tukey’s multiple comparison test to find means that are significantly different from each other between monitored groups.
Reviewer: 26-29 lines. Maybe these lines are confused to read. The authors could use short sentences to express what happened with biochemical parameters and other for chemical composition. For example, Calcium and phosphorus were higher (p<0.05) in rabbits feed with humic substances. Lipid parameters were similar (p>0.05) between the control and HFS group.
Response: We used short sentences to express of biochemical parameters and for description others chemical composition.
Materials and methods
Reviewer: Firstly, if its possible to add ethics statements for this study.
Response: We add ethic statements for this study: The experiment was performed after approval from the Ethics Commission of University of Veterinary Medicine and Pharmacy in Košice (protocol code EKV/2022-11).
Reviewer: 66-73 lines, its important indicates that feed was prepared according a guideline rabbits’ nutritional requirements and indicates chemical composition of the diet or at least calculated chemical composition.
Response: We add sentences for feed preparation according a guideline rabbits’ nutritional requirements and we add into text the evaluation of chemical composition of the diet.
Reviewer: 79-79 lines, this section or paragraph is not according to subtitle. Usually that description is for productive performance not for slaughter.
Response: We have moved the description of fattening and slaughtering to previsious subtitle.
Reviewer: 81 line. What voltage and amperes were used? Was used an equipment? What brand?
Response: We add equipment and parameters for subsequent killing.
Reviewer: 86-88 lines. Is this a proximate analysis?
Response: We described physicochemical analyses of meat samples in methods (line 95-105).
Reviewer: 98-101 lines. The authors need to review their statistical analysis.
Response: We changed statistical method for Tukey’s multiple comparison test to find means that are significantly different from each other between monitored groups.
Results and discussion
Reviewer: 109 line. Table 1. Authors reported daily gain and carcass value, but in methodology is not described that these variables were calculated from data of weight obtained. Also, carcass value is not a term usually used in productive performance parameters. Its most utilized dressing percentage. They need to identify what is the meaning of 35, 60, 90 and 120 days. Average daily gain units maybe they need to review, usually is used g.d-1.
Response: We presented individualy in table 1 the performance and carcass parameters. Also we describe carcass weight and carcass yield in methods.
Reviewer: 105-107 lines. These results are unclear. Statistical analysis needs to review, especially comparison test.
Response: We used Tukey’s multiple comparison test for all results in tables 1 and 2 or figure 1.
Reviewer: 128 line. Is Table 2 indicating a proximal analysis of the meat? review if is water or moisture.
Response: We corrected moisture in table 2.
Conclusion
Reviewer: 154-164 lines. Review conclusion because seems a new discussion or results.
Response: Knowledge in the conclusion results from the statistical evaluation of the results in tables and graphs.

Round 2
Reviewer 2 Report
I think manuscript was improve.